# Interferon-gamma promotes iron export in human macrophages to limit intracellular bacterial replication

**Rodrigo Abreu, Lauren Essler, Pramod Giri, Frederick Quinn** *

Department of Infectious Diseases, University of Georgia, Athens, Georgia, United States of America

* fquinn@uga.edu

**Data Availability Statement:** All relevant data are within the manuscript and its Supporting Information files.

**Funding:** This work was supported in part by an endowment from the University of Georgia Athletic

## Abstract

Salmonellosis and listeriosis together accounted for more than one third of foodborne illnesses in the United States and almost half the hospitalizations for gastrointestinal diseases in 2018 while tuberculosis afflicted over 10 million people worldwide causing almost 2 million deaths. Regardless of the intrinsic virulence differences among *Listeria monocytogenes*, *Salmonella enterica* and *Mycobacterium tuberculosis*, these intracellular pathogens share the ability to survive and persist inside the macrophage and other cells and thrive in iron rich environments. Interferon-gamma (IFN-γ) is a central cytokine in host defense against intracellular pathogens and has been shown to promote iron export in macrophages. We hypothesize that IFN-γ decreases iron availability to intracellular pathogens consequently limiting replication in these cells. In this study, we show that IFN-γ regulates the expression of iron-related proteins hepcidin, ferroportin, and ferritin to induce iron export from macrophages. *Listeria monocytogenes*, *S. enterica*, and *M. tuberculosis* infections significantly induce iron sequestration in human macrophages. In contrast, IFN-γ significantly reduces hepcidin secretion in *S. enterica* and *M. tuberculosis* infected macrophages. Similarly, IFN-γ-activated macrophages express higher ferroportin levels than untreated controls even after infection with *L. monocytogenes* bacilli; bacterial infection greatly down-regulates ferroportin expression. Collectively, IFN-γ significantly inhibits pathogen-associated intracellular iron sequestration in macrophages and consequently retards the growth of intracellular bacterial pathogens by decreasing iron availability.

## Introduction

In Europe and the US alone, more than three million people live with HIV [1–4] and 15 to 19% of the population is over 65-years old [5, 6]. Thus, these approximately 50 million people have weakened immune systems and increased risk of serious complications upon infection with self-resolving pathogens such as *L. monocytogenes* or *S. enterica* [7]. Salmonellosis accounts for 38% of all foodborne diseases in the US and the second most commonly reported gastrointestinal infection in Europe [8–10]. Listeriosis reports are less common but have the highest rates of hospitalization and death among all foodborne illness cases [11, 12]. Tuberculosis is leading cause of death in HIV infected people and is the deadliest infectious disease in

Association to FDQ and a Fulbright PhD
Scholarship to RA (11/278). No additional external
funding was received for this study. The funders
had no role in study design, data collection and
analysis, decision to publish, or preparation of the
manuscript.

**Competing interests:** The authors have declared
that no competing interests exist.

the world on its own [13, 14]. In Europe, almost 60,000 new cases of tuberculosis were
reported in 2018 [11, 15], while in the US, almost 8,000 people were afflicted with this disease
in 2018 [16].

Despite the overall physiological, pathogenic and genetic differences among *L. monocytogenes*, *S. enterica* and *M. tuberculosis*, these pathogens share the ability to survive and replicate
inside non-activated macrophages [17]. By inhibiting macrophage antimicrobial functions,
these pathogens evade both innate and adaptive immune responses and persist within the host
for long periods of time [18]. Furthermore, these three pathogens are associated with reactivation and recurrent infection in immunocompromised individuals such as the elderly or HIV
infected patients [19–23].

IFN-γ is a critical cytokine for innate and adaptive immune responses against intracellular
bacteria [24, 25]. IFN-γ knock-out mice are very susceptible to *L. monocytogenes* [26], *S. enterica* [27] and *M. tuberculosis* [28] infections. In humans, impaired IFN-γ signaling is particularly associated with increased risk of tuberculosis [29]. During the adaptive immune
response, IFN-γ controls the differentiation $CD4_{Th1}$ effector T cells that mediate cellular
immunity against intracellular bacterial infections. Interferon-gamma-activated macrophages
possess up-regulated antigen presentation and increased phagocytosis capabilities, enhanced
production of superoxide radicals, nitric oxide and hydrogen peroxide, and enhanced secretion of pro-inflammatory cytokines [18, 30]. Recently, IFN-γ also has been shown to increase
ferroportin expression in *S. enterica* infected murine macrophages, promoting iron export and
limiting intracellular bacterial replication [31].

Aside from the ability to survive and persist inside non-activated macrophages and other
non-phagocytic cells, *L. monocytogenes*, *S. enterica*, and *M. tuberculosis* also share the ability to
thrive in iron rich environments [32–35]. Deletion of iron acquisition genes in these siderophilic bacteria results in severely attenuated bacterial strains [35, 36], while host iron dysregulation is greatly associated with worsened disease outcomes with all three of these pathogens
[37].

In this study we show that IFN-γ promotes iron export and efficiently prevents pathogen-associated intracellular iron sequestration in THP-1 human macrophages during infections
with *L. monocytogenes*, *S. enterica* and *M. tuberculosis* or the attenuated vaccine strain, *Mycobacterium bovis* BCG. Furthermore, the resulting decrease in intracellular iron availability to
these siderophilic bacteria significantly limits bacterial replication inside the macrophage
resembling the effect of iron chelation therapy. The outcome of this work reveals a novel
mechanism by which IFN-γ limits intracellular bacterial replication in human macrophages.

## Results

### IFN-γ treatment favors iron export in human macrophages

IFN-γ has been previously shown to decrease intracellular iron levels and limit *Salmonella* bacterial replication in mouse macrophages [38]. To determine if IFN-γ can also modulate iron
regulating genes in human macrophages, human THP-1 differentiated macrophages were
treated with human recombinant IFN-γ (200U/ml) and transcriptional expression of the genes
for the iron regulator hepcidin (*HAMP*), iron exporter ferroportin (*SLC40A1*) and intracellular
iron storage protein ferritin (*FTH*) was quantified by qRT-PCR. In agreement with the above-mentioned study, *SLC40A1* transcriptional levels were 2.5-fold higher (±0.23, *p* = 0.005) 16
hours after IFN-γ treatment compared to untreated controls (Fig 1A). Alternatively, transcriptional expression of *HAMP*, (hepcidin down-regulates ferroportin post-translationally)
decreased approximately 70% (*p*<0.001) after IFN-γ treatment (Fig 1B), again biasing towards
an iron export phenotype. These transcriptional data are further supported by the

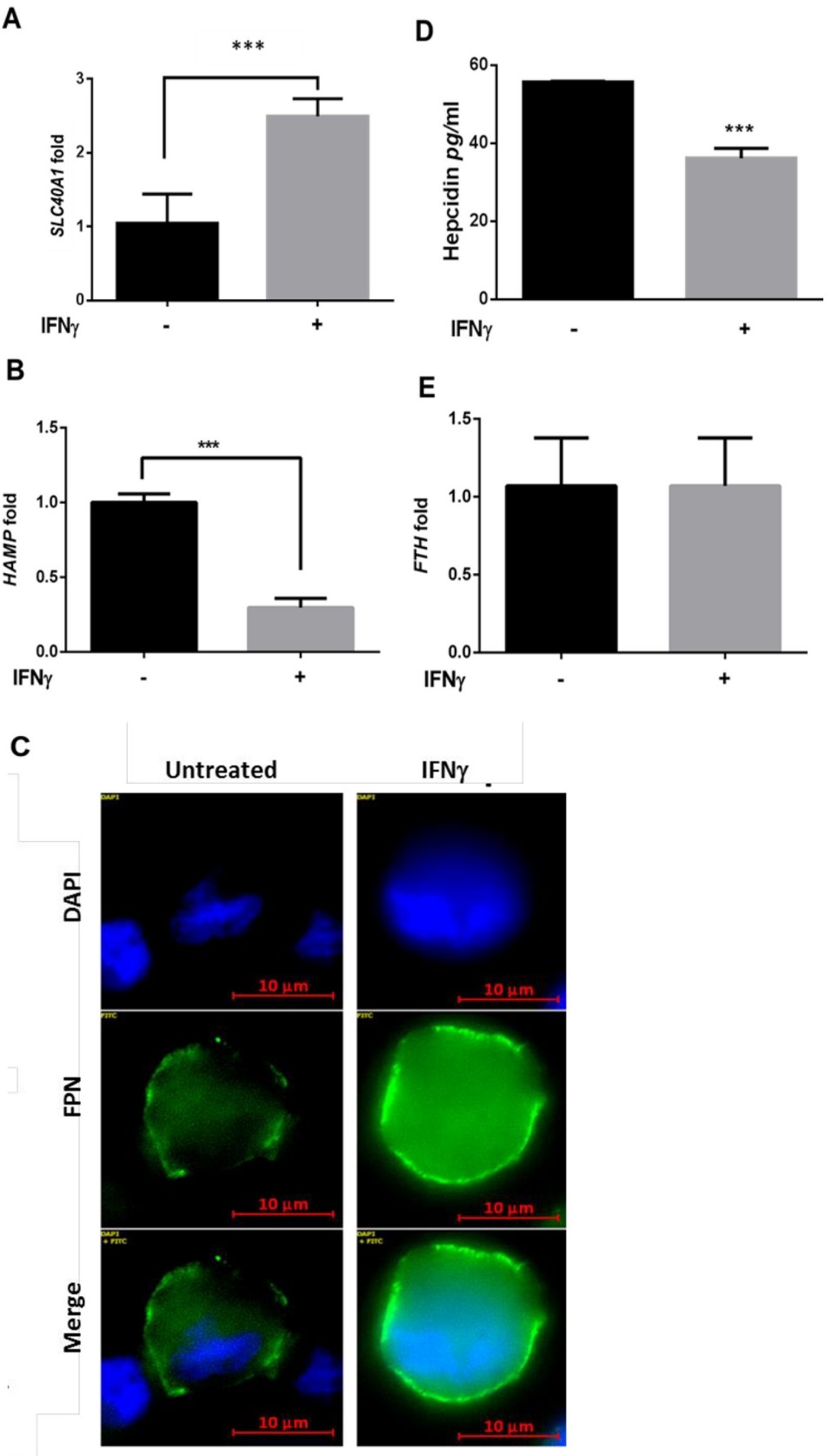

**Fig 1. Interferon-gamma regulates iron-related genes to favor iron export.** A) transcriptional expression levels of *SLC40A1*, and (B) *HAMP* genes in THP-1 macrophages treated overnight with 200U/ml IFN-γ measured by qRT-PCR and compared to untreated controls. C) Ferroportin protein expression assessed in THP-1 macrophages treated as in A. D) Hepcidin secretion levels in THP-1 macrophages treated as in B (magnification = 63X). E) Transcriptional expression levels of the *FTH* gene in THP-1 macrophages treated as in A and B. $^{**}p{<}0.01$, $^{***}p{<}0.001$. All data were from three independent experiments.

corresponding protein levels assayed using immunofluorescence in the culture medium (Fig 1C and 1D). Hepcidin secretion was significantly decreased (Fig 1D, $p{<}0.001$) after IFN-γ treatment while surface ferroportin was increased (Fig 1C). Interestingly, despite the difference in ferroportin and hepcidin expression levels, IFN-γ treatment does not alter transcriptional levels of *FTH* (Fig 1E).

## Siderophilic bacteria manipulate host iron-related proteins to favor intracellular iron sequestration

Intracellular siderophilic bacteria such as *M. tuberculosis*, *M. bovis* BCG, *L. monocytogenes* or *S. enterica* prominently activate Toll-like receptor signaling [25]. Interaction and activation of Toll-like receptors expressed by macrophages induce intracellular iron sequestration both through increased hepcidin secretion and decreased ferroportin expression [39]. To test if these siderophilic organisms can manipulate host iron-related proteins in the macrophage, THP-1 differentiated macrophages were infected with *M. tuberculosis*, *M. bovis* BCG, *L. monocytogenes*, or *S. enterica*, and hepcidin secretion was quantified by ELISA at the peaks of infection (24 hours for *M. tuberculosis* and *M. bovis* BCG, eight hours for *L. monocytogenes* and 16 hours for *S. enterica*). Upon infection, both *M. tuberculosis* and *M. bovis* BCG infected macrophages secreted significantly more hepcidin than respective uninfected controls, 48 hours and 24 hours after infection, respectively (mean difference was 88.6±2.8 pg/ml and 76.2±1.2 pg/ml, respectively, $p{<}0.0001$). This represents an approximate three-fold increase. In the same way, infection with *S. enterica* resulted in increased hepcidin secretion, in agreement with our previous report suggesting that TLR-4 activation is responsible for hepcidin expression in macrophages during infection [39] (Fig 2A).

Alternatively, *L. monocytogenes* infection had no impact on hepcidin secretion (Fig 2A), although it did result in direct ferroportin gene down-regulation, independent of hepcidin expression (Fig 2B and 2C). *Listeria monocytogenes* infected macrophages express lower levels of surface ferroportin compared to uninfected controls (Fig 2B), a 60% decrease measured by mean fluorescence intensity. To confirm that infection with *L. monocytogenes* bacteria down-regulates ferroportin through a hepcidin-independent mechanism, we silenced hepcidin expression through hepcidin gene specific lentiviral ShRNA (S1 Fig). Scramble negative controls (ShScram) and lentiviral ShRNA-silenced expression of the hepcidin gene (ShHAMP) in THP-1 differentiated macrophages were infected with *L. monocytogenes* bacilli; these infected cells expressed similar surface ferroportin levels (Fig 2C and S2 Fig) supporting the hypothesis that this pathogen can promote intracellular iron sequestration, through direct ferroportin down-regulation.

## Pathogen-associated intracellular iron sequestration promotes intracellular replication

Intracellular bacterial pathogens modulate macrophage iron-related proteins to favor iron sequestration (Fig 2). We subsequently determined if *M. bovis* BCG, *L. monocytogenes* or *S. enterica* infected macrophages have increased iron content compared to uninfected controls

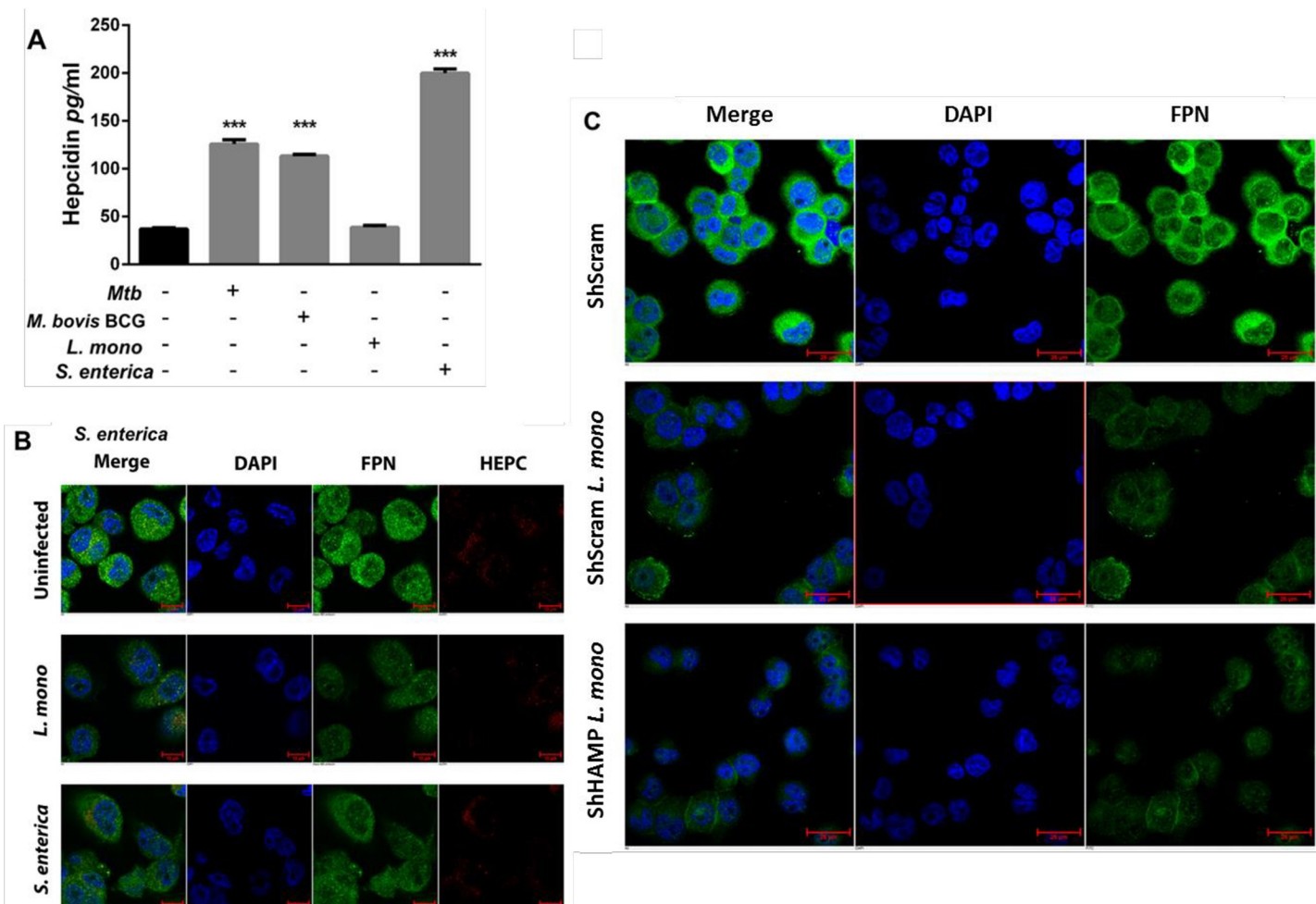

**Fig 2. Intracellular pathogens modulate iron-related proteins to favor intracellular iron sequestration in macrophages.** A) Hepcidin secretion from THP-1 macrophages after infection with *M. tuberculosis* (24 hours), *M. bovis* BCG (24 hours), *L monocytogenes* (eight hours) or *S. enterica* (16 hours) bacilli. B) Ferroportin levels in THP- 1 macrophages eight hours post-infection with *L. monocytogenes* (magnification = 40X). C) Ferroportin expression in hepcidin silenced THP-1 macrophages eight hours post infection with *L. monocytogenes*. Hepcidin gene silencing in THP-1 cells was achieved by lentiviral based shRNA transduction and Scramble short hairpin RNAs (ShScram) were used as a negative control (magnification = 40X). ***$p<0.001$. All data were from three independent experiments.

by Prussian Blue iron staining. Uninfected activated macrophages had low iron retention with minimal iron staining (Fig 3A). However, upon infection with any of the above-mentioned siderophilic bacteria, macrophages had increased intracellular iron levels as observed by increased blue granules (Fig 3B–3D). This observation was reversible with the addition of IFN-γ. When macrophages were activated with IFN-γ before infection with *M. bovis* BCG, *L. monocytogenes* or *S. enterica*, intracellular iron levels significantly decreased, resembling those of uninfected cells (Fig 3E–3G). Interestingly, infections with the three bacterial strains generated different iron staining patterns: *L. monocytogenes* and *M. bovis* BCG infected macrophages had increased intracellular iron levels, but in a similar pattern as uninfected cells (small blue granules dispersed in the cytoplasm) (Fig 3A–3C), while S. *enterica* infected macrophages generated large iron stained granules in the cytoplasm (Fig 3D).

Iron dysregulation is associated with a poorer disease outcome upon infection with siderophilic bacteria such as *M. tuberculosis*, *L. monocytogenes* and *S. enterica* [40]. Alternatively, iron chelation has proven to be an effective therapy *in vitro* and *in vivo* against some of these

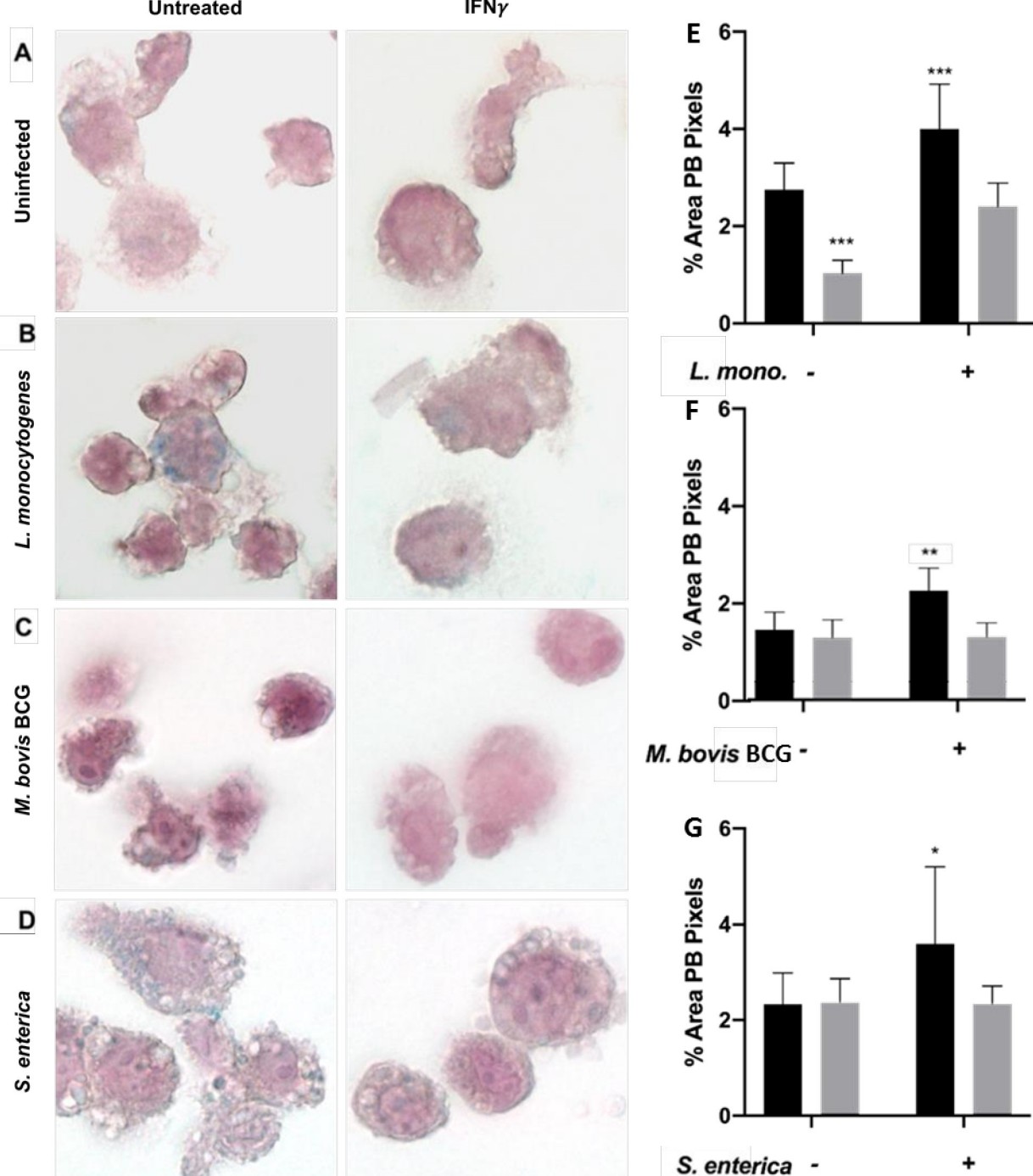

**Fig 3. Interferon-gamma treatment decreases pathogen-associated intracellular iron sequestration.** A) Intracellular iron levels assessed by Prussian Blue staining in untreated and IFN-γ activated THP-1 macrophages, and (B-D) infected with three siderophilic bacteria (magnification = 100X). Percentage total area of Prussian Blue (PB) pixels in macrophages after infection with (E) *L. monocytogenes*, (F) *M. bovis* BCG, or (G) *S. enterica* with or without IFN-γ treatment. *p<0.05, **p<0.01, ***p<0.0001.

siderophilic pathogens [41]. To evaluate if increased intracellular iron sequestration was essential for bacterial replication, we infected THP-1 differentiated macrophages with *M. tuberculosis*, *L. monocytogenes* and *S. enterica* bacteria in the presence of the iron chelators

deferoxamine (DFO) or deferiprone (DFP) and intracellular replication was assessed using the gentamicin protection assay. DFO was chosen for *M. tuberculosis* infection as an injectable chelator which has been previously validated for use with *Mycobacteria* [42], while DFP was used for *L. monocytogenes* and *S. enterica* infections as this oral human therapeutic chelator is more physiologically relevant for gastrointestinal pathogens. As expected, iron chelation significantly decreased intracellular replication of *M. tuberculosis* (90% less 72 hours post-infection, S3C Fig), *L monocytogenes* (72% less eight hours post-infection, S3D Fig) and *S. enterica* (89% less 16 hours post-infection, S3E Fig).

## Interferon-gamma prevents pathogen induced iron modulation in macrophages

Interferon-gamma treatment and infection with various intracellular pathogens have opposing effects on macrophage survival (Figs 1 and 2). Thus, IFN-γ treatment was assessed for its ability to prevent iron retention in macrophages infected with *M. bovis* BCG, *L. monocytogenes* or *S. enterica* bacteria. THP-1 activated macrophages were treated with 200U/ml human recombinant IFN-γ overnight and infected with the various species of intracellular bacteria. At different time points after infection, ferroportin levels were assessed by immunofluorescence and hepcidin secretion by ELISA. Similar to what was observed with uninfected macrophages (Fig 1), IFN-γ treatment increased ferroportin expression in THP-1 macrophages infected with *L. monocytogenes* (Fig 4A), *M. bovis* BCG (Fig 4B), and *S. enterica* (Fig 4C). MFI quantification revealed that IFN-γ treatment significantly increased ferroportin expression by 60% and 74% for *L. monocytogenes* and *S. enterica*, respectively (S4 Fig).

Additionally, IFN-γ also decreases hepcidin secretion in the culture supernatants of *L. monocytogenes* (mean difference 37.9±1.5 pg/ml) and *M. bovis* BCG (mean difference 50.9±1.5 pg/ml) infected macrophages at eight and 24 hours post-infection, respectively (Fig 4D and 4E). Surprisingly, IFN-γ only marginally inhibits hepcidin secretion form *S. enterica* infected macrophages to levels still significantly higher than uninfected controls (mean difference 47.2 ±1.97 pg/ml, *p*<0.001) (Fig 4F).

## Interferon-gamma limits intracellular *Salmonella* and *Mycobacterium* bacterial replication in macrophages through hepcidin inhibition

Treatment with IFN-γ counteracts pathogen modulation of iron-regulating genes (Fig 4) favoring iron export. It has been previously reported that IFN-γ limits iron availability to intracellular pathogens through up-regulation of ferroportin leading to decreased bacterial replication [31]. To assess if hepcidin inhibition and increased ferroportin expression would translate to decreased intracellular bacterial replication, ShHAMP THP-1 macrophages were infected and intracellular bacterial replication assessed using the gentamicin protection assay. As observed with IFN-γ treatment, hepcidin silencing leads to increased ferroportin expression in uninfected macrophages favoring iron export (Fig 1B). Upon infection with *S. enterica*, ShHAMP THP-1 macrophages showed significantly decreased intracellular bacterial replication than respective negative scramble controls (ShScram) (90% decrease) at 16 hours post infection (Fig 5A). A similar impact (70% decrease) in intracellular bacterial replication was observed at 48 and 72 hours post infection with *M. tuberculosis* (Fig 5B). However, *L. monocytogenes* replication was not altered in ShHAMP THP-1 macrophages, suggesting that *Listeria*-mediated iron sequestration is hepcidin-independent (Fig 5C).

Interferon-gamma limits intracellular bacterial replication in macrophages though activation of multiple anti-microbial mechanisms [43]. To confirm that the concentrations tested in this work inducing iron export also reduced intracellular bacterial replication, we treated

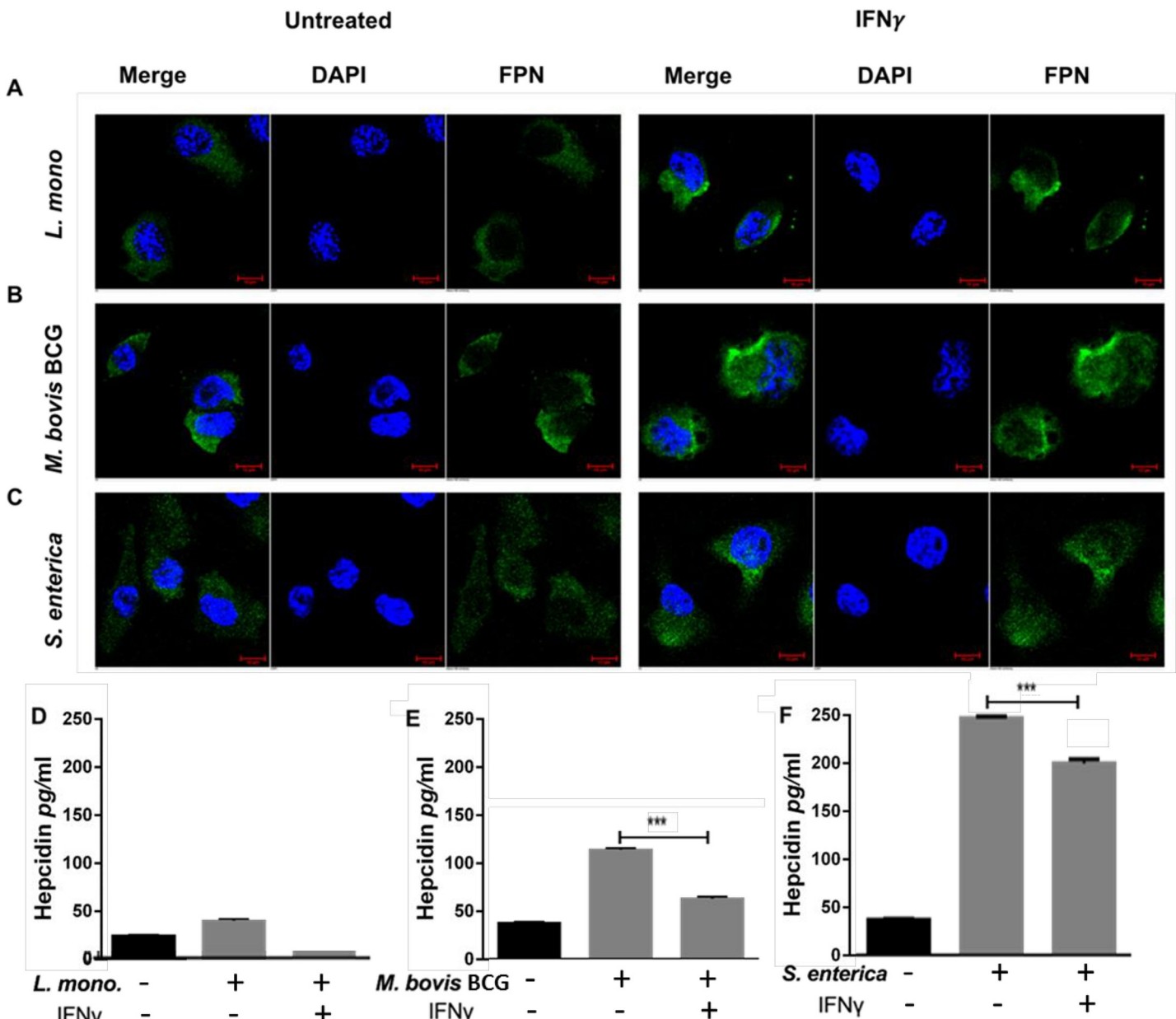

**Fig 4. Interferon-gamma prevents pathogen associated iron modulation in macrophages.** A) Ferroportin expression in IFN-γ-activated (200U/ml) macrophages eight hours post infection with *L. monocytogenes*, (B) 48 hours post infection with *M. bovis* BCG, or (C) 16 hours post infection with S. *enterica* (magnification = 40X). D) Hepcidin secretion in the culture supernatants of IFN-γ-activated macrophages eight hours post infection with *L. monocytogenes*, (E) 24 hours post infection with *M. bovis* BCG, or (F) 16 hours post infection with S. *enterica*. \*\**p*<0.01, \*\*\**p*<0.001. All data were from three independent experiments.

THP-1 differentiated macrophages with 200U/ml IFN-γ before infection with *L. monocytogenes*, *S. enterica* or *M. tuberculosis* and quantified intracellular bacterial burden in a gentamicin protection assay. *L. monocytogenes*, *S. enterica* and *M. tuberculosis* infected macrophages had significantly decreased intracellular bacterial burdens after IFN-γ treatment at 8, 16 and 24 hours post infections, respectively (Fig 5D–5F). In *S. enterica* or *M. tuberculosis*-infected macrophages, IFN-γ has a significant impact on intracellular bacterial counts 16 and 48 hours post-infection, where it translates into a 90% decrease in bacterial numbers compared to

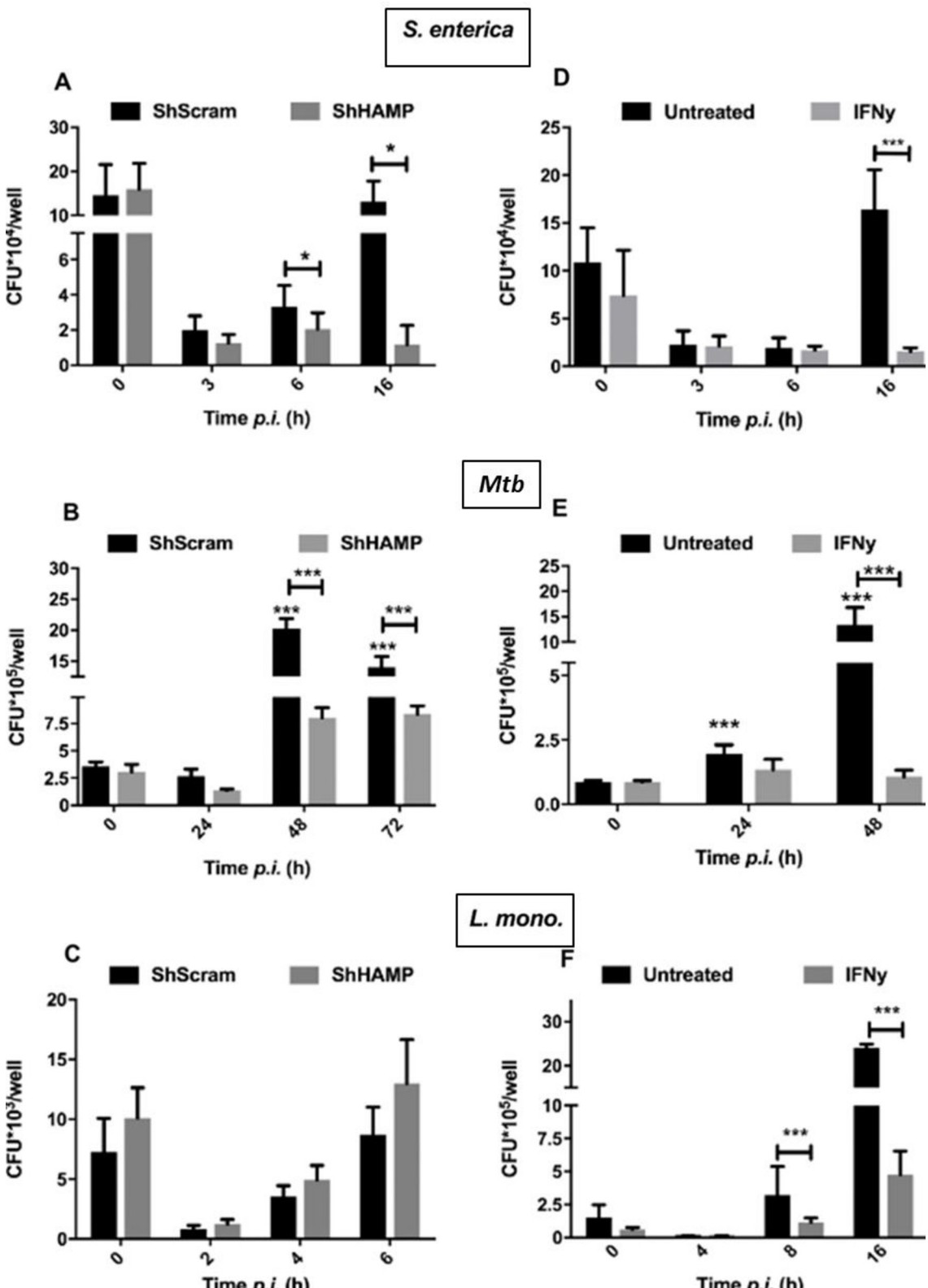

**Fig 5. Hepcidin inhibition limits intracellular _M. tuberculosis_ and _S. enterica_ bacterial replication in macrophages.** _Salmonella enterica_ (A), _M. tuberculosis_ (B) and _L. monocytogenes_ (C) intracellular burden in ShHAMP THP-1 macrophages and respective ShScram controls. _Salmonella enterica_ (D), _M. tuberculosis_ (E) and _L. monocytogenes_ (F) intracellular burden in IFN-γ-activated macrophages (200U/ml) and respective untreated controls. *$p < 0.05$, ***$p < 0.001$. All data were from three independent experiments.

untreated controls (Fig 5D and 5E). Similarly, IFN-γ treatment results in an 80% decrease in *L. monocytogenes* intracellular bacterial numbers six hours post-infection (Fig 5F).

## Discussion

The host immune response to intracellular bacteria is a complex network of pro- and anti-inflammatory mediators assuring efficient bacterial killing with minimal tissue damage [18]. In contrast, bacterial persistence is a fine tuning of virulence factors that coordinate bacterial survival within the host with minimal activation of the surveillance immune system [17]. *Mycobacterium tuberculosis*, *L. monocytogenes* and *S. enterica* serovar Typhimurium are three intracellular bacterial species that persist in macrophages and efficiently avoid the host immune system. Despite the varied factors involved in bacterial survival and replication inside the macrophage, these three pathogens share the ability to avoid or inhibit macrophage anti-microbial functions such as phagosome maturation, phagolysosome fusion or induction of nitric and oxygen reactive species [17]. In contrast, IFN-γ macrophage activation promotes intracellular bacterial killing through direct induction of the abovementioned antimicrobial mechanisms [44]. In this study, we describe a novel mechanism by which IFN-γ limits intracellular bacterial replication in macrophages. In human macrophages, IFN-γ promotes iron export and efficiently prevents pathogen-associated intracellular iron sequestration. The consequent decrease in intracellular iron availability to these siderophilic bacterial pathogens significantly limits their replication inside the macrophage.

Iron is a crucial micronutrient to all forms of life with important biological functions. This metal is a component of molecules involved in sensing, transporting and storing oxygen, and of enzymes involved in oxidation and reduction of substrates during energy production, intermediate metabolism, and the generation of reactive oxygen or nitrogen species for host defense. During infection with siderophilic bacteria, decreased iron availability greatly impedes intracellular bacterial replication [40]. Pathogen-associated intracellular iron regulation in macrophages is dependent on TLR signaling and mediated through two independent and redundant mechanisms regulating the iron-related proteins hepcidin and ferroportin [39]. Secreted hepcidin binds to surface ferroportin of mammalian cells to induce its internalization and degradation, resulting in decreased iron export [45]. While TLR-4, TLR-7 and TLR-8 signaling induces hepcidin secretion, TLR-1, TLR-2 and TLR-6 activation significantly inhibits ferroportin expression in THP-1 human macrophages [39].

In this study we show that *L. monocytogenes* infection promotes intracellular iron sequestration in macrophages through ferroportin downregulation, independent of hepcidin expression (Fig 2). These results are consistent with predominant TLR-2 activation during *L. monocytogenes* infection [46], and in support of previously published results [47] where *L. monocytogenes* significantly decreases ferroportin expression through a hepcidin-independent mechanism. IFN-γ treatment significantly increases ferroportin expression in THP-1 macrophages even after *L. monocytogenes* infection (Fig 3A) inhibiting *Listeria*-associated intracellular iron sequestration.

Increased hepcidin expression has been shown to promote intracellular *M. bovis* BCG replication [39, 48], and HIV replication is augmented in hepcidin-treated macrophages [49]. *Salmonella enterica* significantly induces hepcidin expression consistent with TLR-4 signaling (Fig 2), whereas *M. tuberculosis* and *M. bovis* BCG, which activate both TLR-2 and TLR-4, promote intracellular iron sequestration through hepcidin-independent and dependent mechanisms (Fig 3). In this study, it was observed that IFN-γ treatment inhibits hepcidin secretion in human macrophages (Fig 1) and significantly decreases pathogen-induced hepcidin secretion during *M. bovis* BCG or *S. enterica* infection (Fig 4), subsequently reducing intracellular iron sequestration in infected macrophages (Fig 3).

*M. bovis* BCG was used as model for *M. tuberculosis* in several assays that required analysis at biosafety level 2 (Figs 3 and 4). However, we feel *M. bovis* BCG is an adequate substitute for these studies and the data generated here are comparable to *M. tuberculosis* for a number of reasons: It is well known that *M. bovis* BCG maintains a high genetic homology and similar cell wall composition to *M. tuberculosis* [50, 51], and it has been observed here (Fig 1) and in other studies [52–54] that within the 24–48 hour infection times used in this study, no differences in the bacterial replication rates, trafficking patterns, or host cell viability rates are observed between *M. tuberculosis* and *M. bovis* BCG infected cell lines. These common traits at least in the earliest stages of infection allow *M. bovis* BCG to extrinsically activate TLR signaling and induce hepcidin secretion and intracellular iron retention much like *M. tuberculosis* (Figs 2 and 3). All of these common traits are important when considering their contributions to maximizing vaccine efficacy for *M. bovis* BCG.

Ferroportin overexpression in murine macrophages is able to severely impair *S. enterica* growth [55]. Similarly, reducing hepcidin gene expression in ShHAMP THP-1 cells reduces *S. enterica* replication, showing that IFN-γ-mediated hepcidin down-regulation alone can significantly impact intracellular replication (Fig 5). Moreover, sodium phenylbutyrate, a strong hepcidin inhibitor in macrophages (unpublished data) has been shown to significantly inhibit *S. enterica* growth *in vivo* [56]. In contrast, *L. monocytogenes* intracellular bacterial burden remains unaltered in ShHAMP macrophages, indicating that IFN-γ-induced ferroportin expression is an important factor limiting bacterial growth during *L. monocytogenes* infection (Fig 5).

Bacteria possess a myriad of mechanisms to scavenge the host iron pool, and the three pathogens used in this study utilize different iron scavenging strategies [57]. *Mycobacterium tuberculosis* siderophores, mycobactin and carboxymycobactin, efficiently recruit and scavenge iron in the phagosome [58]. Carboxymycobactin is the major iron-chelator for both free and protein-bound iron in the macrophage phagosome and cytoplasm [58, 59], while surface mycobactin acts as a membrane chelator and iron-transporter recovering iron from carboxymycobactin and host ferritin [60]. In macrophages, ferritin mostly localizes to the nucleus with minimal cytoplasmic distribution [61], and *M. bovis* BCG-infected macrophages present increased iron retention in the nucleus with some diffuse iron distribution in the cytoplasm (Fig 3). Iron-loaded ferritin has been previously shown to be efficiently recruited to the phagosome and utilized by *M. tuberculosis* [60]. Future studies may explore the impact of IFN-γ in intracellular iron distribution within the macrophage and its accessibility to mycobacteria.

*Salmonella enterica* inhibits phagolysosome fusion, and persists and replicates inside the immature phagosome compartment [62]. During infection, efficient control of intra-phagosome iron levels by the phagosomal iron exporter NRAMP is essential to limit bacterial replication [38, 63]. *Salmonella enterica* iron acquisition strategies are very similar to other Gram-negative bacteria and mostly dependent on the ferric siderophores enterochelin and salmochelin [34, 64]. These siderophores scavenge iron from the host proteins transferrin and lactoferrin, with the iron-laden siderophores then transported through bacterial outer-membrane receptors IroN and FepA [64, 65]. Besides this mechanism, S. *enterica* also can utilize heme-iron sources inside the phagosome, although this seems to be more prominent during infection of hemophagocytic macrophages [65]. Consistent with the use of intra-phagosomal iron sources, S. *enterica* infected macrophages have localized iron aggregates (Fig 3) possibly associated with immature phagosomes where the bacteria persist. Although iron supplementation decreases bacterial survival during early stages of infection, probably through increased ROS generation, at 16 hours post infection increased iron levels are detrimental for the host and facilitate bacterial replication (S5 Fig). This supports the hypothesis of pathogen-mediated iron recruitment and accumulation in the phagosome. This accumulated iron then counteracts

NRAMP iron export from the phagosome at later stages of infection which is needed for efficient bacterial clearance. Aside from confirming iron localization to the phagosome, future studies may assess how iron gets recruited to this compartment and how NRAMP impacts iron distribution in the macrophage.

Siderophore synthesis genes are absent in the *L. monocytogenes* genome, therefore heme-bound iron is proposed as the major iron source utilized by this pathogen during macrophage infection [66]. Phagosomal activation of the pore-forming protein listeriolysin-O leads to bacterial escape from the phagosome to the cytoplasm [24, 67]. Once in the cytoplasm expression of the ferrochrome ABC transporters *hupCGD (lmo2429/30/31)* enhances iron acquisition from heme-proteins [33, 66]. The diffuse cytoplasmic distribution of intracellular iron in *L. monocytogenes* infected macrophages (Fig 3) may represent an increase in heme-proteins which can be efficiently used as an iron source. In the future it would be interesting to identify the major heme-proteins targeted by *L. monocytogenes* for iron scavenging and assess the impact of IFN-γ signaling on the expression of these same proteins.

Hepcidin was first identified as an antimicrobial peptide utilized by the host cell during infection with extracellular pathogens; this work has been extensively reported [68, 69]. As with lactoferrin, hepcidin efficiently decreases extracellular iron availability to pathogens such as *Vibrio cholerae* [63, 70]. However, during infection with intracellular pathogens hepcidin-mediated intracellular iron sequestration in macrophages is deleterious for the host and facilitates bacterial replication [49, 55, 71, 72]. Furthermore, hepcidin has been reported to play an anti-inflammatory role during chronic infections which could further dampen an effective immune repose against persistent intracellular pathogens [73, 74].

## Conclusions

Interferon-γ is an important cytokine in both the innate and adaptive immune responses against intracellular pathogens. This cytokine upregulates major histocompatibility complex class I and class II antigen presentation, and contributes to macrophage activation by increasing phagocytosis and priming the production of pro-inflammatory cytokines and potent antimicrobials, including superoxide radicals, nitric oxide, and hydrogen peroxide [43]. Interferon-gamma also controls the differentiation CD4$_{Th1}$ effector T cells which mediate cellular immunity against intracellular bacterial infections. The role of IFN-γ in regulating intracellular iron availability for *S. enterica* has been previously reported, but with conflicting results. While IFN-γ-mediated nitric oxide production significantly increased ferroportin expression in murine macrophages which significantly contributed to limiting intracellular bacterial replication [43], IFN-γ treatment has also been shown to upregulated hepcidin expression in a murine macrophage *M. tuberculosis* infection model [75]. This report contradicts our observations that describe a positive outcome where IFN-γ strongly promotes iron export in human macrophages through increased ferroportin expression and decreased hepcidin secretion. The consequent decrease in intracellular iron availability severely limits replication of three different bacterial pathogens, *L. monocytogenes*, *S. enterica* and *M. tuberculosis*. Thus, our study elucidates a novel mechanism by which IFN-γ controls intracellular bacterial replication and exposes iron dysregulation as an important factor of innate immunity against these pathogens.

## Materials and methods

### Cell culture and macrophage differentiation

The THP-1 monocytic cell line was obtained from ATCC (#TIB-202) and maintained in complete RPMI with 2mM glutamine and supplemented with 10% heat inactivated fetal bovine serum (C-RPMI). For differentiation into a macrophage-like phenotype, cells were resuspended

at a concentration of $8X10^5$ per ml, treated with 50nM phorbol 12-mytistate 13-acetate (PMA) for 24 hours and rested overnight in C-RPMI with 100μM ferric ammonium citrate (FeAC) and 200U/ml of human recombinant IFN-γ (R&D Systems, MN USA) unless otherwise stated.

## Hepcidin silencing

The THP-1 monocytic cell line was transduced with gene-silencing Short-hairpin RNA (ShRNA) lentiviral particles (Santa Cruz Biotech, TX, USA). Briefly, $2 X 10^3$ THP-1 cells were grown in v-bottom 96-well plates with Hepcidin-specific ShRNA lentiviral particles or respective scramble control at a multiplicity of infection of 10 with 5 μg/ml polybrene. Cells were centrifuged at 900x$g$ for 30 minutes to increase contact and incubated overnight at 37˚C with 5% $CO_2$. Cells were then centrifuged for 5 minutes at 400x$g$, resuspended in C-RPMI, monitored for viability and sequentially expanded to 48-wells in C-RPMI. When monolayers reached 50% confluency in 48 well plates, stably transduced cells were selected with 1 μg/ml puromycin and expanded in T75 flasks, before storage in liquid nitrogen. Thawed aliquots were passaged once before selection with puromycin.

## Bacterial strains and infection

The strains used in this study were *M. bovis* BCG (Pasteur), and *M. tuberculosis* (Erdman) kindly provided by Dr. Jeffery Cox (UC Berkley, CA, USA). *Listeria monocytogenes* was acquired from ATCC (#15313; VA USA) and clinical isolate *Salmonella enterica* serovar Typhimurium was kindly provided by Dr. Mary Hondalus (UGA, GA USA). Mycobacteria were grown to an $OD_{600} \approx 0.8$a.u. in Middlebrook 7H9 medium supplemented with Albumin Dextrose Catalase (ADC), 5% glycerol and 0.5% Tween 80. Frozen stocks were prepared in 20% glycerol 7H9 medium (v/v) and maintained at -80˚C. *Listeria monocytogenes* and *S. enterica* were grown to an $OD_{600} \approx 0.8$a.u. in brain-heart infusion (BHI) or Luria-Bertani broth, respectively. Frozen stocks were made in the respective media with 20% glycerol (v/v) and stored at -80˚C. To test viability of the frozen stocks, colony forming units/ml were determined by serial dilution and plating of the thawed suspensions on the respective agar media three weeks after freezing. Before infection, *M. bovis* BCG or *M. tuberculosis* bacilli were passed through a 21G syringe and opsonized for two hours in RPMI with 10% non-heat inactivated horse serum at 37˚C with gentle rocking.

For mycobacterial infections, $3X10^5$ PMA-differentiated THP-1 macrophages were incubated in C-RPMI with opsonized bacilli in 48 well plates. Infections were performed using a multiplicity of infection of five to 10 bacilli per cell, for two hours at 37˚C with 5% $CO_2$. After internalization, macrophages were washed twice with PBS and left on C-RPMI with 50 μg/ml gentamicin and 200 U/ml IFN-γ throughout infection. For intracellular bacterial burden quantification, host cells were lysed at indicated time points with 0.1% TritonX-100 for 10 minutes and serial dilutions plated in 7H10 agar medium. Bacterial colonies were counted twice after 19 to 23 day incubations at 37˚C.

For *L. monocytogenes* and *S. enterica* infections, macrophages were seeded as described above and incubated with non-opsonized bacteria in C-RPMI for one hour at 37˚C with 5% $CO_2$. After internalization, the intracellular bacterial burden was determined as described above for mycobacterial infections but instead using BHI or LB agar plates after 24 hours incubation at 37˚C.

## RNA extraction and real-time PCR

Total cellular RNA from $1X10^6$ THP-1 macrophages was extracted with TRIzol (Invitrogen, Thermo Fisher Scient. MA USA) following the manufacturer's protocol and reverse

transcribed into cDNA using a SuperscriptIII First strand cDNA synthesis Kit (Invitrogen, Thermo Fisher Scientific. MA USA) with poly $dT_{20}$ primers. Quantitative PCR (qPCR) was performed using Bio- Rad IQ SYBR green supermix (Bio-Rad, CA USA) in a iQ™5 Real-Time PCR Detection System. All values were normalized against reference gene *GAPDH* (ΔCT = CT [target]—CT [reference]). Fold change in expression was calculated as $2^{-\Delta\Delta CT}$, where ΔΔCT = ΔCT (test sample)—ΔCT (control). The primer sequences for the genes examined were the following: human *HAMP*, forward, 5 = -GGATGCCCATGTTCCAGAG-3 =; reverse, 5 = -AGCACATCCCACACTTTGAT-3 =; human *GAPDH*, forward, 5 = -GCCCTCAACGACCACTTTGT -3 =; reverse, 5 = -TGGTGGTCCAGGGGTCTTAC- 3 =, human *SLC40A1*, forward, 5 = -CACAACCGCCAGAGAGGATG-3 =; reverse, 5 = -ACCAGAAACACAGACACCGC-3 =; Human *FTH*, forward, 5 = -AGAACTACCACCAGGACTCA-3 =; reverse, 5 = -TCATCGCGGTCAAAGTAGTAAG-3 =.

## Hepcidin secretion quantification

Hepcidin levels in culture supernatants were determined using human hepcidin DuoSet ELISA Kit (R&D Systems, MN, USA), per manufacture's recommendations.

## Immunofluorescence microscopy

Anti ferroportin and anti-hepcidin antibodies for ferroportin and hepcidin detection were kindly provided by Dr. Tara Arvedson, and immunofluorescence staining was performed as previously described [76]. Briefly, $2X10^5$ THP-1 macrophages were grown and differentiated in eight or 16 well chamber microscopy slides and infected as described above, fixed with 4% paraformaldehyde (PFA), and permeabilized with 0.1% Triton X-100. For ferroportin staining, cells were incubated with 2 μg/ml mouse antibody diluted in C-RPMI overnight. For detection, cells were incubated with 2 μg/ml goat anti-mouse alexa-fluor-488 (Invitrogen, Thermo Fisher Scient. MA USA) at 4˚C for two hours. Cells were counterstained with DAPI. For hepcidin staining, cells were infected, fixed and permeabilized as described above, and stained with 2 μg/ml mouse anti-hepcidin antibody overnight at 4˚C. Slides were imaged in a Zeiss Axiovert 200M microscope at 40X and 63X and images acquired with Axiocam MRm grey scale camera.

## Prussian Blue for iron staining

THP-1 macrophages were grown to $4X10^5$ cells per well in 8 well chamber microscopy slides and differentiated as described above. After infection, cells were fixed with 4% formaldehyde in PBS for 10 minutes at room temperature, washed with PBS and stained twice with a 4% hydrochloric acid and 4% potassium ferrocyanide (1:1 v/v) solution of for 25 minutes (Polysciences Prussian Blue stain kit, PA USA). After washing with PBS, cells were counterstained with filtered 1% Nuclear Fast red solution for 5 to 10 minutes, washed gently with PBS and water, and mounted and imaged using an Olympus Bx41 microscope. Images were acquired with an Olympos DP71 color camera using 20X, 40X and 100X lenses, and processed with cellSens v1.14.

## Image analysis

Image analysis and mean pixel fluorescence intensity were determined with Zeiss Axiovision Rel 4.8.1 software. Co-localization and Prussian Blue staining were quantified with image J 1.51K software. Grey scale images were converted to binary files for automatic shape analysis. Protein-protein co-localization was determined by double positive pixel areas.

Prussian Blue staining was quantified in 20x color image thresholds for background and determined as percentage of blue pixel area over total pixel area averaged from at least four different fields from three independent experiments.

## Statistics

All data are presented as means ± SD. Statistical significance differences between groups were determined using Student's *t* test with GraphPad Prism software (CA, USA).

## Supporting information

**S1 Fig. Hepcidin silencing in THP-1 macrophages.** A) Hepcidin secretion in ShHAMP THP-1 macrophages and respective ShScram controls after infection with *M. bovis* BCG, *L. monocytogenes* and *S. enterica*. B) Surface ferroportin expression in ShHAMP THP-1 macrophages and respective ShScram controls measured by flow cytometry.
(TIF)

**S2 Fig. *Listeria monocytogenes* downregulates ferroportin by a hepcidin-independent mechanism.** Ferroportin expression in ShHAMP THP-1 macrophages eight hours post-infection with *L. monocytogenes* and 16 hours post-infection with *S. enterica*. Ferroportin levels were quantified by mean fluorescence intensity of 40 cells from three different fields of three independent experiments. ***$p < 0.001$.
(TIF)

**S3 Fig. Iron chelation inhibits intracellular bacterial replication.** A) Intracellular iron Prussian Blue staining in macrophages infected with three siderophilic bacteria. B) Percentage of Prussian Blue (PB) pixels in THP-1 macrophages after infection with three siderophilic bacteria. (C) *Mycobacterium tuberculosis* intracellular burden in THP-1 macrophages in presence of iron chelator DFO. D) *Listeria monocytogenes* intracellular burden in THP-1 macrophages in presence of iron chelator DFP. E) *Salmonella enterica* intracellular burden in THP-1 macrophages in presence of iron chelator DFP. **$p < 0.01$, ***$p < 0.001$. All data were from three independent experiments.
(TIF)

**S4 Fig. Interferon-gamma induces ferroportin expression after *Listeria monocytogenes* bacterial infection.** Ferroportin in IFN-γ-treated THP-1 macrophages eight hours post-infection with *L. monoytogenes*, 16 hours post-infection with *S. enterica* and 24 hours post-infection with *M. bovis* BCG bacteria. Ferroportin levels were quantified by mean fluorescence intensity of 40 cells from three different fields of three independent experiments. *$p < 0.05$, ***$p < 0.001$.
(TIF)

**S5 Fig. Iron impacts intracellular replication of siderophilic bacteria in macrophages.** A) THP-1 macrophages differentiated as described in Materials and Methods, rested and infected in iron-supplemented medium. *Listeria monocytogenes* (A) and *S. enterica* (B) intracellular bacterial burdens were determined by a gentamicin protection assay. ***$p < 0.001$. Data were from three independent experiments.
(TIF)

**S6 Fig. Hepcidin silencing decreases intracellular *Mycobacterium bovis* BCG replication.** *Mycobacterium bovis* BCG intracellular burden in ShHAMP THP-1 macrophages 24 hours post-infection. **$p < 0.01$. Data were from three independent experiments.
(TIF)

## Acknowledgments

We thank Dr. Tara Arvedson (Amgen Inc., CA, USA) for providing ferroportin monoclonal antibodies. We also thank Shelly Helms for technical assistance in several parts of this project.

## Author Contributions

**Conceptualization:** Rodrigo Abreu, Pramod Giri.

**Data curation:** Rodrigo Abreu, Lauren Essler.

**Formal analysis:** Rodrigo Abreu, Lauren Essler, Pramod Giri.

**Funding acquisition:** Frederick Quinn.

**Investigation:** Rodrigo Abreu, Lauren Essler, Pramod Giri.

**Methodology:** Rodrigo Abreu, Pramod Giri.

**Project administration:** Pramod Giri, Frederick Quinn.

**Resources:** Pramod Giri, Frederick Quinn.

**Supervision:** Frederick Quinn.

**Validation:** Rodrigo Abreu, Lauren Essler.

**Visualization:** Rodrigo Abreu, Lauren Essler.

**Writing – original draft:** Rodrigo Abreu.

**Writing – review & editing:** Rodrigo Abreu, Lauren Essler, Pramod Giri, Frederick Quinn.

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
