## [Decision Letter · Decision Letter 0]

9 Nov 2020

Interferon-gamma promotes iron export in human macrophages to limit intracellular bacterial replication

PONE-D-20-30583

Dear Dr. Quinn,

We’re pleased to inform you that your manuscript has been judged scientifically suitable for publication and will be formally accepted for publication once it meets all outstanding technical requirements.

Kind regards,

Martin E Rottenberg

Academic Editor

PLOS ONE

Journal Requirements:

1.Thank you for stating the following financial disclosure:

 [The funders had no role in study design, data collection and analysis, decision to publish, or preparation of the manuscript.].

Please provide an amended Funding Statement that declares *all* the funding or sources of support received during this specific study (whether external or internal to your organization) as detailed online in our guide for authors at http://journals.plos.org/plosone/s/submit-now. 

Please state what role the funders took in the study.  If any authors received a salary from any of your funders, please state which authors and which funder. If the funders had no role, please state: "The funders had no role in study design, data collection and analysis, decision to publish, or preparation of the manuscript."

Please send your amended statements by return email; we will change the online submission form on your behalf.

Reviewers' comments:

Reviewer's Responses to Questions

**Comments to the Author**

1. Is the manuscript technically sound, and do the data support the conclusions?

Reviewer #1: Yes

2. Has the statistical analysis been performed appropriately and rigorously? 

Reviewer #1: Yes

3. Have the authors made all data underlying the findings in their manuscript fully available?

Reviewer #1: Yes

4. Is the manuscript presented in an intelligible fashion and written in standard English?

Reviewer #1: Yes

5. Review Comments to the Author

Reviewer #1: Dr. Abreu and colleagues have comprehensively revised their manuscript entitled ‘Interferon-gamma promotes iron export in human macrophages to limit intracellular bacterial replication’ (No. PONE-D-20-30583). In doing so, they have appropriately responded to the comments of all five reviewers. The authors have substantially improved their manuscript which should be of interest for a broad readership of ‘PLoS One’.

6. PLOS authors have the option to publish the peer review history of their article (what does this mean?). If published, this will include your full peer review and any attached files.

Reviewer #1: **Yes: **Manfred Nairz

---

## [Editor Report · Acceptance letter]

25 Nov 2020

PONE-D-20-30583 

Interferon-gamma promotes iron export in human macrophages to limit intracellular bacterial replication 

Dear Dr. Quinn:

I'm pleased to inform you that your manuscript has been deemed suitable for publication in PLOS ONE. Congratulations! Your manuscript is now with our production department. 

Kind regards, 

on behalf of

Dr. Martin E Rottenberg 

Academic Editor

PLOS ONE